# Gender Gap in Mental Health during the COVID-19 Pandemic in South Korea: A Decomposition Analysis

**DOI:** 10.3390/ijerph20032250

**Published:** 2023-01-27

**Authors:** Sunoong Hwang, Heeju Shin

**Affiliations:** 1Department of Economics, Pukyong National University, Busan 48513, Republic of Korea; shwang@pknu.ac.kr; 2Department of Sociology, The Catholic University of Korea, Bucheon 16442, Republic of Korea

**Keywords:** COVID-19, mental health, gender gap, labor market

## Abstract

The economic and social effects of the COVID-19 pandemic have been widespread but unevenly distributed among genders. The pandemic may have also affected men’s and women’s mental health differently. This study examined whether the pandemic had stronger adverse effects on women’s mental health than on that of men given that the decline of the labor market was greater for women than for men. Using data from South Korea (June/September/December 2020, *N* = 3000), we investigated the gender gap in mental health during the first year of the COVID-19 pandemic and its association with gender differences in labor market experiences. We employ the Blinder–Oaxaca decomposition method for this analysis. Although depression and anxiety increased among employed women and men during COVID-19, women showed lower levels of mental health than men. A significant portion of this gender gap is explained by women experiencing greater job loss, income reduction, and prohibition of remote work than men. We also find that women in their 30s experienced greater mental health problems than men of the same age even after controlling for other conditions. Overall, our findings show that a greater proportion of employed women than men experienced poor labor market conditions and increased family burdens during the COVID-19 pandemic, which contributed to women reporting worse mental health than men.

## 1. Introduction

Since the first case was reported in December 2019, COVID-19 has rapidly altered the lives of almost every person on the planet in significant ways. Far from the optimistic, early 2020 projection that the virus would disappear soon, people are still living through the COVID-19 pandemic in 2023. The psychological challenges imposed by the prolonged pandemic have become important social and academic issues. A growing body of literature is documenting the negative influence of the pandemic on mental health, focusing on worsened economic situations and changes in work and family life as mediating factors [1,2,3,4].

Studies on the association between the COVID-19 pandemic and people’s mental health have shown that women’s mental health has been more affected by the COVID-19 pandemic than men’s [5,6,7,8]. These studies have documented that social and economic disadvantages, such as job loss, financial hardship, and rearrangements of household roles, have been concentrated on women, causing women to experience more psychological distress than men, even though the pandemic has had widespread effects on everyone’s lives regardless of gender. This development of gender gaps in labor market outcomes and mental health during this pandemic differs substantially from past economic crises for the following reasons.

First, men were typically more likely than women to be laid off and less likely to be employed during past recessions [9,10,11]. For instance, during the global financial crisis of 2008–2009, the number of employed persons aged 15–64 in OECD member countries fell by 2.6% for men compared to a 0.7% decline for women (Figure 1a). This gender difference in labor market disadvantages during previous recessions has been largely explained by the ways that occupational distribution varies by gender: men are more likely to be employed in construction and manufacturing, whose production cycles are affected by macroeconomic shocks on a large scale, whereas women tend to be employed in less cyclical positions, such as service industries and public administration [10,11,12].

Second, although it is well documented that women generally have worse mental health than men [13,14,15,16], men are more likely than women to be affected psychologically by economic deterioration such as job loss or lowered job quality [17,18,19]. The gender gap in the effects of economic factors on mental health suggests differences in the social and economic roles played by men and women. Men are more likely to be attached to the labor market emotionally and physically and to have greater psychosocial needs for being employed than women because of their traditional family responsibilities as the main income earner [17,20].

As a result of these factors, past economic crises have often had a greater negative impact on men’s mental health than women’s [20,21,22] (however, there is ongoing debate on this issue; see [22]). In contrast, the economic and social effects of the COVID-19 pandemic have different gendered mechanisms from typical economic recessions. Studies show that the labor market consequences of COVID-19 have been reversed in terms of gender from those of previous recessions [9,11,23,24,25]. Between 2019 and 2020, the number of employed women aged 15–64 decreased by 3.7% in OECD economies, while the same rate of decline was 3.0% for men (Figure 1b). According to statistics of the International Labour Organization (https://ilostat.ilo.org/, accessed on 15 November, 2022), which included a total of 98 countries, the rate of decrease in world employment in 2020 was 3.0% and 4.1% for men and women, respectively. This larger decrease in women’s employment during the COVID-19 pandemic is primarily due to a drastic reduction in the demand for face-to-face services as a result of government-mandated non-pharmaceutical interventions and social distancing policies, as well as the consumer response to the pandemic [9,11,25,26]. In addition, school and daycare closures have forced women to engage in childcare and household chores at a greater level than before the pandemic, which has led to higher stress in women than men [7,8,27,28]. Several studies using datasets released during the COVID-19 pandemic display a negative association between time spent engaging in childcare involved in homeschooling and subjective well-being [29,30].

In this paper, we examine how the effects of COVID-19 on mental health varied by gender using data from South Korea collected during the first year of the pandemic. We also investigate key factors explaining this gender gap in mental health by applying the Blinder–Oaxaca decomposition method. While there now exists a sizeable body of evidence showing the gendered effects of the COVID-19 pandemic on mental health, a systematic investigation of potential causes of this differential impact remains scarce, and more evidence is needed to identify the nature of the pandemic’s effects on people’s lives and to suggest effective work and family policies. The present paper aims to fill this deficiency.

South Korea is particularly suitable for our analysis for two reasons. First, it was one of the first countries in the world to experience a large number of confirmed COVID-19 cases and for which government-mandated restrictions on economic and social activities were quickly implemented. Second, the gender gap in employment is larger and traditional gender role attitudes are stronger than those of other OECD countries; women work disproportionately more in service and sales jobs and spend much more time on household care than men. It is postulated that these differences worsened women’s mental health during the pandemic compared to those of women in countries with a smaller gender disparity. Additionally, we expect that there was a significant age effect on this impact since women’s burden of housework might be greatest in their 30s and 40s when they have young children, while men did not demonstrate such age variation.

This study contributes to the large body of literature on general gender differences in mental health by investigating the impacts of social and economic changes caused by COVID-19. Our formal decomposition analysis distinguishes our study from many recent works on gender inequality in mental health during COVID-19 that did not quantify the relative importance of various potential causes of such consequences. In this regard, most close to our work are the studies of Zoch et al. [31] and Etheridge and Spantig [32], who conducted the same decomposition analysis as ours using data from Germany and the UK, respectively. While those studies found no significant mental health effect of gender in labor market experiences during the first months of the COVID-19 pandemic, we provide opposing evidence, highlighting the importance of labor market channels in South Korea. We also find a significant psychological effect of gender differences in the ability to work from home. Overall, our results show the presence of substantial cross-country variations in the mechanisms through which the COVID-19 pandemic exacerbated gender inequalities in mental health, raising the need for policy responses considering the economic and institutional characteristics of each country.

## 2. Materials and Methods

### 2.1. Data and Measurement

We used data from the ‘COVID-19-Related Survey for Employees’ conducted in South Korea by Embrain at the request of Jikjang-Gapjil 119 (workplace harassment 119), a private organization aimed at public interest. The survey was conducted four times using the same questionnaire scheme in April, June, September, and December of 2020 to track the effects of the COVID-19 pandemic on the working and living conditions of wage workers. A total of 4000 respondents, 1000 per wave, aged 19 to 55 who were employed at the time they were surveyed completed the structured survey questionnaire online. The sample was selected using proportional distribution random sampling based on the proportion of the employed population from the Economically Active Population Survey conducted by Statistics Korea.

From our perspective, the dataset from the survey has two important advantages: timeliness and representativeness. Survey administration started right after the first major spread of COVID-19, and its sample design ensured that the proportions of demographic groups of workers were consistent with those of the Economically Active Population Survey. The survey included questions about respondents’ general sociodemographic characteristics, overall perceptions of COVID-19, and the effects of the pandemic on their work lives. However, the first wave does not include some key variables used in this study, such as job loss experienced by the respondents. Therefore, we analyzed only the 3000 respondents from the 2nd, 3rd, and 4th waves. In our sample of three combined survey waves, each observation belongs to a unique individual.

The key independent variables in this study are changes in respondents’ work lives caused by the COVID-19 pandemic, such as job loss, decreased working hours, decreased income (measured separately for respondents and their household members), and mandatory continued presence at the workplace. Gender is used as a key variable for a pooled model and a variable for group division, which we will explain in the Methods section. A limitation of this study is that we could not examine individuals with non-binary gender identities because the survey we used collected no such information.

Demographic characteristics (age, educational attainment, marital status, and region), occupational factors (firm size, occupation), and work conditions (monthly income, weekly working hours, and job status) were included as control variables.

Our dependent variables are self-reported degrees of depression and anxiety caused by COVID-19. These variables were measured using the questions, “how depressed do you feel due to the COVID-19 pandemic?” and “how much anxiety do you feel due to the COVID-19 pandemic?”. The respondents answered on a 4-point Likert scale, with higher scores indicating more negative feelings. These variables represent respondents’ subjective assessment of their mental health problems and do not indicate psychiatric illness or disorder that involves clinically significant distress and impairment in functioning [4].

Despite the potential issue of validity of our dependent variable due to the measurement problem, we used the current dataset because of its advantage of timeliness. In addition, the dataset allows us to examine the gender effects of pandemic depression specifically rather than general depression by asking about the level of depression due to COVID-19. The data have another important advantage that they were collected across four waves in 2020, allowing analysis over time. We considered using more recent data collected in April 2022, in which depression was measured by the PQH-9. However, the data were collected more than two years after the onset of the pandemic, limiting their applicability to examining the impacts of the pandemic on mental health. Instead, we repeated our analyses using the 2022 dataset to demonstrate the robustness of our study. The results were basically the same as those previously found.

### 2.2. Methods of Analysis

We conducted two types of analysis. First, we examined which factors influenced by COVID-19 were associated with depression and anxiety and looked for gender effects. To do so, we built regression models using depression and anxiety as cardinal variables and estimated the models using ordinary least squares (OLS). Three regression models were analyzed. In a pooled model, we estimated the effects of independent variables on mental health by including all the individuals, and the same analysis was applied to two models constructed separately for women and men to examine how the independent variables act differently in these two groups.

Second, we applied the Blinder–Oaxaca (B-O) decomposition to examine which factors explain the gender difference in mental health. Although it might be more appropriate to estimate ordered logit or probit models for our analyses, we applied OLS regression analyses because detailed decompositions using such nonlinear models are not readily available. We tested the robustness of the results from OLS regressions using ordered probit models and found equivalent results.

We started the B-O decomposition by estimating the following regression separately for men and women:
(1)
$$Y_{i}=X_{i}^{\prime }\beta_{i}+\varepsilon_{i}$$

where 
$Y_{i}$
 refers to the outcome variable for individual 
i
, 
$X_{i}^{\prime }$
 is a vector of observable socioeconomic characteristics, 
$\beta_{i}$
 is a vector of coefficients to be estimated, and 
$\varepsilon_{i}$
 is an error term. The standard B-O decomposition is presented as follows:
(2)
$${\bar{Y}}_{f}-{\bar{Y}}_{m}={\left({\bar{X}}_{f}-{\bar{X}}_{m}\right)}^{\prime }{\hat{\beta }}_{f}+{\bar{X}}^{\prime }{}_{m}\left({\hat{\beta }}_{f}-{\hat{\beta }}_{m}\right)$$

where the subscripts 
f
 and 
m
 stand for female and male, respectively, and the upper bar (
$\bar{A}$
) denotes the mean of the corresponding underlying variable. The left side of this equation indicates the difference between women and men in mental health. The first term on the right side represents the part of the gender difference explained by the difference in means of observable characteristics, and the second term captures the part that is not explained by such mean differences in observable characteristics but can be attributed to differentiated effects of the same characteristics.

This method is used to determine the amount of group difference in outcome variable due to observable personal characteristics compared to that due to structural and unknown factors. As such, the B-O decomposition method has the advantage of not only specifying variables that contribute to the difference, but also identifying the difference explained by structural characteristics which is regarded as discrimination. The B-O decomposition is a special case of the statistical method to quantify the relative contribution of each mediator in 
X
 (e.g., labor market experiences) to the total effect of a focal predictor 
Z
 (e.g., gender) on the outcome variable 
Y
 (e.g., mental health) [33,34]. Unlike the conventional approach in mediation analysis, which adds covariates sequentially into the regression, the B-O decomposition estimates the contributions of all potential mediators in a model at the same time. It does not depend on the assumption that confounders and mediators are not correlated, and its results are not affected by the ordering of variables. For a detailed description of the method the reader is referred to [32,35].

We modified this standard method in two ways. First, because the results of that decomposition could differ depending on the choice of reference group, we followed Neumark [36], whose method was also used by Madden [37] and Etheridge and Spantig [32] to analyze gender differences in mental health. That form of the decomposition is as follows:
(3)
$${\bar{Y}}_{f}-{\bar{Y}}_{m}={\left({\bar{X}}_{f}-{\bar{X}}_{m}\right)}^{\prime }{\hat{\beta }}^{*}+{\bar{X}}^{\prime }{}_{f\;}\left({\hat{\beta }}_{f}-{\hat{\beta }}^{*}\right)+{\bar{X}}^{\prime }{}_{m\;}\left({\hat{\beta }}^{*}-{\hat{\beta }}_{m}\right)$$

where 
${\hat{\beta }}^{*}$
 is a vector of coefficients from the regression equation using the pooled data. The sum of the last two terms on the right side of the equation indicates the part unexplained by gender differences in the mean of the observable characteristics.

Second, we used a method suggested by Oaxaca & Ransom [38] and Jann [35] to resolve the methodological issue that the results of the decomposition might be influenced by the reference category of the explanatory variable when a categorical variable is applied in the model.

Our analyses were carried out using Stata software version 15.0 (StataCorp LLC, Collage Station, TX, USA). The B-O decomposition used its oaxaca command written by Jann [35].

## 3. Results

### 3.1. Descriptive Findings

The mental health changes during the pandemic are shown by gender in Figure 2. Generally, the mental health of both women and men has worsened over time. In women, the depression index was 1.93 in April 2020, three months after the first COVID-19 case was reported in South Korea, and it had increased to 2.28 by December. The increase in depression for men was similar, but men’s overall mental health was better than that of women. Anxiety also increased for both women and men over time and was higher in women than men in general.

Table 1 shows descriptive statistics for the variables examined in this study. The table includes important information about gender differences in the explanatory variables, from which we can infer potential contributors to the gender gap in mental health. Additionally, the differences of group means will be used as an important component for B-O analyses, the results of which we will present later. There are significant differences in selective job characteristics and working conditions between men and women. Women are more concentrated in service and sales jobs and at small firms than men are. In addition, monthly income is lower and weekly working hours are fewer for women than men, and women represent a larger proportion of non-regular workers than men. Additionally, gender differences are noted in labor market experiences in the aftermath of COVID-19. Women experienced job loss, a decrease in working hours and household income (their own and their family members), and a lack of permission to work at home at higher levels than men during 2020, and the results of the *t*-test for these variables are all significant at the 0.05 level. From these facts, we suspect that the greater effects of COVID-19 on the work lives of women might be related to their higher risks for depression and anxiety.

### 3.2. Multivariate Regression Results

For the B-O decomposition, we need the coefficients of the explanatory variables estimated separately for pooled and individual gender groups, in addition to the mean difference of each variable between the two groups. Table 2 shows the results of our OLS analyses to examine the associations between mental health and relevant factors. Geographic regions were included in the analysis, but most of them were found to be statistically insignificant. We do not report these results for brevity.

As shown in the first column of this table using the pooled data, the gender gap in mental health remained significant at the 0.01 level even after controlling for demographic characteristics, job characteristics, and the effects of the pandemic. We confirmed the significant effects of age on mental health. Compared with people in their 50s, those in their 30s experienced greater psychological distress, but other age groups did not. Monthly income has significant effects on depression; the level of depression decreases as income increases. Consistent with the time trajectories observed in Figure 2, we found a significant increasing trend of depression throughout the first year of the COVID-19 pandemic.

Most importantly, the effects of COVID-19 on work life significantly influenced mental health in several ways. Those who experienced job loss during the pandemic had greater depression than those without job loss experience. A reduction in individual or household income during the pandemic was associated with increased depression. Workers who were not allowed to work at home during the pandemic reported greater depression than those who could work from home. However, working hour reduction did not have a significant relationship with depression in the absence of income reduction.

The remaining two columns of Table 2 present the results for subsamples by gender, showing differences in the effects of the same observable characteristics. Most notably, the effect of age on depression was present only for women. Women in their 30s reported the most frequent depression, but we found no age effect for men. In contrast, gender differences in the effects of labor market experiences during the pandemic were not as large as the age effects.

It is notable that variables related to current job characteristics presented significant effects on men’s mental health only. Men working in small- or medium-sized firms reported less depression than those in large or public firms. The negative association between monthly income and depression was significant among men but not among women. Additionally, although labor market influences of COVID-19 had significant effects on both men’s and women’s mental health, men were affected greatly by not being allowed to work from home compared with women.

The results for anxiety are similar to those for depression. Thus, for brevity, we present these results in Appendix A
Table A1. In addition, we conducted the same regression analyses using models without weekly working hours to assess its mediating effect. However, there was no significant difference in the results, indicating no notable mediating effect of weekly working hours. Additionally, although weekly working hours are significantly correlated with some key variables used in our study, a VIF test did not show an indication of multicollinearity.

### 3.3. Decomposition of the Gender Gap in Mental Health

As shown in Table 3, the gender gap in the average depression index was approximately 0.23. The explained portion of the group difference was 0.06, which accounts for 26.5% of the actual difference. In other words, if women had the same characteristics as men, the difference in average depression index between the groups would decrease by 26.5%, and only the unexplained difference would remain.

The breakdown shows that the biggest contribution to the gender gap is monthly income, which accounts for about 73% (=0.0442/0.0602) of the total explained gender difference in depression. This is because, as seen in Table 1 and Table 2, women earn significantly less money than men (−), and income level is negatively related with depression or anxiety (−), which together positively and significantly contribute to gender difference of mental health. The next largest contribution is job loss (22.7%). Women experienced job loss at a greater proportion than men (+), and job loss was positively associated with depression (+). The third contributing factor is income reduction of a family member (19.8%), followed by prohibited from working from home (18.4%) and personal income reduction (18.2%). All these labor market experiences during the COVID-19 pandemic were also more frequent among women than men (Table 1) and had a positive association with depression (Table 2). These findings indicate that gender differences in the frequencies of these variables positively contribute to a gender gap in depression. The decomposition of the gender gap in anxiety had similar findings. The results are provided in Appendix A
Table A2.

The unexplained portion in the B-O decomposition indicates that the gap remains even when controlling for the mean differences between women and men and is thus caused by unobservable factors. As shown in Table 3, people in their 30s make a significant contribution to the unexplained portion of the gender gap in depression. In other words, women in their 30s experienced greater depression than men of the same age, and age affects mental health unequally between men and women when other conditions are held the same.

Although it is not simple to estimate why the effects of age on mental health vary by gender among people in their 30s in this study, we assume that working women in their 30s are more likely than men to suffer from an increased burden of childcare and household chores due to school closures and labor market changes during the pandemic. To further investigate the effects of the pandemic on married women, we decomposed the gender gap in depression by marital status. We found that age did not make a significant contribution to the unexplained portion of the gender gap in the not-married sample, but age in the 30s had a significant contribution in the married sample. These findings do not directly explain the coefficient effect of the gender gap in mental health. However, they do support the idea that married women with young children have suffered psychologically from the effects of COVID-19 on their work and family lives much more than have married men of the same age.

## 4. Discussion

Generally, labor market deterioration has negative effects on mental health [39,40]. The unemployed are threatened by a bad economic climate because of fewer opportunities for job searches and reemployment [41,42], and currently employed workers are also negatively affected because they can face job vulnerability in an insecure labor market [40]. However, the effects of any economic crisis on well-being are not distributed evenly among genders [43,44].

The results of our study support previous studies showing that women workers experienced more frequent job losses, decreases in work hours, decreases in household income, and being forbidden to work from home than men did during the first year of COVID-19. Although our regression analysis provides evidence that those who experienced being laid-off, decreases in their own income, and not being allowed to work remotely during the pandemic were more likely than others to have lower mental health, it did not identify which factors are relevant to the gender gap in mental health. Our B-O decomposition revealed that labor market experiences differentiated by gender during the pandemic made a significant contribution to the gender difference we found in depression. A greater proportion of employed women than men experienced job loss, decreased income, and were not allowed to work remotely, and those differences all contributed to the lower mental health reported by women.

In contrast, our results contradict those reported in works by Zoch et al. [31] and Etheridge and Spantig [32], who applied the same decomposition analysis as ours to data collected from Germany and the UK, respectively, in the first months of the COVID-19 pandemic. Those two studies reached the same conclusion that, while decreases in mental health were more pronounced for women than for men, they could not be explained by gender differences in changes in working conditions due to the pandemic. Instead, both studies found that a significant part of the gender gap in mental health could be explained by larger increases in loneliness among women than men. Figure 3 suggests why our results are different from those studies. Decreases in employment during the COVID-19 pandemic were smaller for women than for men in Germany and the UK, whereas this decline was more than 1.5 times larger for women compared to men in South Korea. This indicates that the gendered labor market impact of COVID-19 substantially varied across countries, as did the mechanism through which COVID-19 affected mental health by gender. Nevertheless, as can be inferred from Figure 1, it seems more likely that the labor market channel around the world widened the gender gap in mental health.

It is also noteworthy that in our decomposition results, a sizable portion of the unexplained gender gap in depression can be attributed to the coefficient effect of being in one’s 30s. In studies of income inequalities among gender groups, the unexplained portion indicates structured inequalities that are not explained by differences in observed characteristics. Although it is not intuitive to interpret the unexplained part of the gender gap in depression, this also might reflect the effects of structured inequality on depression by gender.

In other words, just being a woman in her 30s is significantly associated with a higher level of depression than being a woman of a different age and being a man of any age. Therefore, we assume that men and women in their 30s have different structural life conditions than people of other ages, and that these differences generally damage women’s well-being. Additionally, the finding that the coefficient effect of being in one’s 30s was significant only for married women, not for unmarried women, suggests that the structural life condition harming the mental health of women is closely related to their gender roles within their families.

As documented by previous researchers, working mothers spend more time than working fathers caring for their children. In addition to financial hardship and job losses concentrated on women as a result of the pandemic, increased care responsibilities and household chores due to daycare and school closures seem to have affected the mental health of working mothers more negatively than working fathers during the pandemic crisis [5,6,7,8,28,29,30].

Additionally, the varied effects of current-job-related factors on mental health by gender need to be discussed. From the results of regression analyses, current job factors of firm size, monthly income, and weekly working hours have significant mental health effects on men only, while age has an effect on women only. These findings are consistent with our discussion earlier in this paper that men’s mental health is more likely to be affected by economic factors than women’s is because men tend to be more involved in the labor market as the main income earners. This finding is also related with women being more likely influenced by family-related components as a main caregiver for family members.

## 5. Conclusions

Rearrangements of childcare provisions, household chores, and employment to cope with a crisis reflect the bargaining power of household members, including transaction costs, and conflicts among family members often arise during the negotiation process [45]. Thus, it is worrisome that the prolonged pandemic or its recurrence could be structuralizing unequal positions between men and women within households and influencing the future career prospects of women, which could have long-term, negative effects on their mental health [25].

Our study suggests that any labor market structure that perpetuates gender inequality in society should be restructured, and that social systems and policies reproducing gender inequalities should be improved. Working women are more frequently exposed to time-pressure regarding work-family conflicts and social norms. This affects women’s mental and physical health, and these effects were exacerbated during the prolonged pandemic crisis. The adverse effects of the pandemic on women’s lives will also have negative influences on the well-being of their families and communities. For these reasons, community and nationwide efforts are needed to eliminate the detrimental effects of the pandemic on women.

Although many countries are starting to return to pre-pandemic activities, the global economy remains strained. South Korea has been considered to respond better to the COVID-19 pandemic than other high-income countries in terms of the number of deaths, rate of vaccination, etc. However, it is uncertain whether Korean society will also recover from the pandemic faster than other countries. Despite the recent increase in gender role flexibility in the labor market and within families, gender issues persist in many areas, and family-friendly social systems are not well established in South Korea. As long as the current structural discrimination persists, the higher social cost of women will continue.

This study has several limitations. Although we did not focus on how the effects of COVID-19 varied with the socioeconomic characteristics of women, some of our findings indicate that the pandemic has not affected women in a homogenous way. The negative effects of COVID-19 have concentrated on women with socioeconomic disadvantages, such as those working irregular or part-time jobs, those with a low educational attainment, and those with a low income. Further research is needed on how COVID-19 affects women’s mental health differently by social class. Another limitation is that our variables of mental health may lack robustness due to measurement strategy. Although our data have advantage of being collected using questions specifically about the influences of COVID-19, more appropriate measures of mental health would have improved this study. It is also necessary to examine whether the mechanisms emphasized in this study contributed to gender disparities in objectively measurable psychiatric disorders.

Despite these limitations, this study is meaningful in that it analyzed impacts on the labor market and mental health in the early days of a pandemic using timely collected data.

## Figures and Tables

**Figure 1 ijerph-20-02250-f001:**
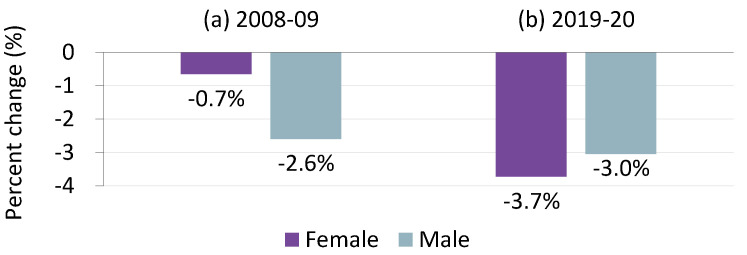
Percent change in OECD total employment by gender during the global financial crisis (2008−2009) and the COVID-19 pandemic (2019−2020). Notes: Workers aged 15–64; data from OECD.Stat, available online: https://stats.oecd.org (accessed 15 November 2022).

**Figure 2 ijerph-20-02250-f002:**
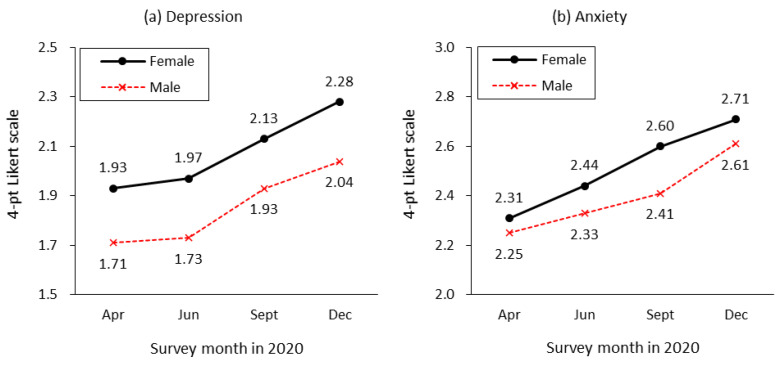
Mental health by gender during the first year of the COVID-19 pandemic. Notes: Depression and anxiety were measured using a 4-point Likert scale ranging from 1 (not at all) to 4 (very severe).

**Figure 3 ijerph-20-02250-f003:**
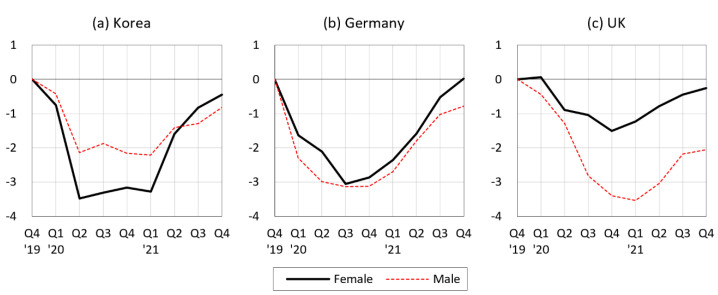
Percent change in employment by gender during the quarters of the first two years of the COVID-19 pandemic compared to 2019 Q4: a comparison between South Korea, Germany, and the UK. Notes: Workers aged 15–64, seasonally adjusted, data from OECD.Stat, available online: https://stats.oecd.org (accessed on 15 November 2022).

**Table 1 ijerph-20-02250-t001:** Descriptive statistics.

Variable	Total (N = 3000)		Women (N = 1289)		Men (N = 1711)		Gender Difference	
Depression (4-point Likert scale)	1.998	(0.791)	2.127	(0.772)	1.900	(0.792)	0.227	*
Anxiety (4-point Likert scale)	2.509	(0.740)	2.586	(0.741)	2.451	(0.735)	0.135	*
Married, %	48.6		46.6		50.1		−3.5	
Age 20s, %	20.6		20.7		20.5		0.2	
Age 30s, %	28.9		28.5		29.2		−0.7	
Age 40s, %	33.4		33.8		33.0		0.8	
Age 50s, %	17.1		16.9		17.3		−0.4	
High school, %	20.7		23.9		18.4		5.5	*
College-3 years, %	20.1		22.3		18.4		3.9	*
College-4 years, %	51.1		48.2		53.3		−5.1	*
Graduate, %	8.1		5.6		10.0		−4.4	*
Professional, %	50.0		47.6		51.8		−4.2	*
Service/sales, %	27.5		37.5		19.9		17.6	*
Engineer, %	10.2		3.6		15.2		−11.6	*
Manual labor/etc., %	12.3		11.4		13.0		−1.6	
Large firm/public firm, %	30.7		27.1		33.5		−6.4	*
Medium-sized firm, %	30.4		26.8		33.2		−6.4	*
Small firm, %	38.1		45.3		32.7		12.6	*
Other, %	0.7		0.9		0.6		0.3	
Monthly income (mil KRW, monthly)	2.765	(1.580)	2.085	(1.031)	3.275	(1.722)	−1.190	*
Weekly working hours	40.55	(12.87)	36.77	(12.89)	43.40	(12.09)	−6.600	*
Non-regular worker, %	40.0		48.6		33.5		15.1	*
Job loss experience, %	15.1		17.9		12.9		5.0	*
Decrease in working hours, %	27.5		33.0		23.4		9.6	*
Income decrease (own), %	33.1		36.4		30.6		5.8	*
Income decrease (family members), %	41.3		48.5		35.9		12.6	*
Work from home not allowed, %	14.4		16.4		13.0		3.4	*

Notes: Continuous variables are presented as mean (standard deviation) and categorical variables as percentage (%). The population distribution across regions is not included in this table for brevity. Asterisks (*) represent significance of *t*-tests for gender differences at the 0.05 level.

**Table 2 ijerph-20-02250-t002:** Results of OLS regressions for depression.

Variables (Reference)		Total			Women			Men		
		Coef.		S.E.	Coef.		S.E.	Coef.		S.E.
Woman		0.167	***	(0.031)						
Married		0.045		(0.032)	−0.011		(0.047)	0.090	*	(0.047)
AGE(50s)	Age 20s	0.054		(0.051)	0.113		(0.071)	−0.002		(0.073)
	Age 30s	0.095	**	(0.043)	0.189	***	(0.063)	0.017		(0.060)
	Age 40s	0.042		(0.038)	0.063		(0.056)	0.017		(0.051)
Education (high school)	College-3 years	0.008		(0.045)	0.001		(0.065)	0.007		(0.063)
	College-4 years	0.037		(0.039)	0.035		(0.055)	0.020		(0.054)
	Graduate	0.130	**	(0.061)	0.093		(0.099)	0.102		(0.080)
Job (professional)	Service/sales	0.049		(0.040)	0.105	*	(0.056)	−0.020		(0.058)
	Engineer	−0.014		(0.053)	0.140		(0.122)	−0.063		(0.062)
	Manual labor/etc.	0.057		(0.053)	0.088		(0.076)	0.040		(0.073)
Firm (large/public)	Medium	−0.034		(0.036)	0.081		(0.056)	−0.118	**	(0.048)
	Small	−0.120	***	(0.036)	−0.074		(0.052)	−0.157	***	(0.051)
	Other	−0.140		(0.162)	−0.049		(0.266)	−0.237		(0.168)
Occupational characteristics	Monthly income	−0.037	***	(0.012)	−0.028		(0.026)	−0.056	***	(0.014)
	Weekly working hours	0.002		(0.001)	0.001		(0.002)	0.003	*	(0.002)
	Non-regular worker	−0.032		(0.038)	0.016		(0.057)	−0.071		(0.053)
Influence of COVID-19	Job loss experience	0.273	***	(0.048)	0.283	***	(0.068)	0.275	***	(0.066)
	Decrease in working hours	−0.016		(0.039)	0.004		(0.060)	−0.034		(0.052)
	Income decrease (own)	0.189	***	(0.049)	0.165	**	(0.073)	0.193	***	(0.065)
	Income decrease (family members)	0.095	**	(0.039)	0.074		(0.054)	0.107	*	(0.057)
	Work from home not allowed	0.327	***	(0.041)	0.252	***	(0.057)	0.398	***	(0.058)
Survey	3rd wave	0.187	***	(0.033)	0.187	***	(0.050)	0.200	***	(0.044)
(2nd wave)	4th wave	0.293	***	(0.034)	0.343	***	(0.052)	0.267	***	(0.046)
Constant		1.761	***	(0.096)	1.657	***	(0.145)	1.717	***	(0.139)
Observations		3000			1289			1711		
Adjusted R-squared		0.131			0.123			0.111		

Notes: Effects of region were estimated but are not included in this table. *** *p* < 0.01, ** *p* < 0.05, * *p* < 0.1.

**Table 3 ijerph-20-02250-t003:** Blinder-Oaxaca decomposition of the gender gap in depression.

	Total (N = 3000)				Married (N = 1459)				Not Married (N = 1541)			
Women	2.1272				2.0815				2.1672			
Men	1.9001				1.9079				1.8921			
Difference	0.2272				0.1736				0.2750			
Explained	0.0602		26.50%		0.0766		44.10%		0.0486		17.70%	
Unexplained	0.1669		73.50%		0.0970		55.90%		0.2264		82.30%	
	Explained		Unexplained		Explained		Unexplained		Explained		Unexplained	
Married	−0.0016		−0.0487		-		-		-		-	
Age 20s	0.0000		0.0066		−0.0007		−0.0053		0.0003		0.0204	
Age 30s	−0.0003		0.0256	*	0.0000		0.0513	**	−0.0012		0.0123	
Age 40s	0.0000		−0.0126		−0.0003		0.0187		−0.0006		−0.0141	
Age 50s	0.0002		−0.0142		0.0015		−0.0086		0.0001		−0.0024	
High school	−0.0024		0.0006		−0.0064		−0.0073		0.0007		0.0054	
College-3 years	−0.0014		−0.0010		−0.0048		0.0102		0.0000		−0.0070	
College-4 years	0.0003		0.0075		−0.0007		0.0113		−0.0001		0.0017	
Graduate	−0.0038	**	0.0002		−0.0098	**	−0.0007		0.0004		0.0003	
Professional	0.0010		−0.0461		0.0047		−0.0188		−0.0024		−0.0601	*
Service/sales	0.0045		0.0055		0.0001		−0.0107		0.0042		0.0215	
Engineer	0.0043		0.0056		0.0041		0.0030		0.0048		0.0108	
Manual labor/etc.	−0.0006		−0.0055		0.0043		0.0006		−0.0024		−0.0110	
Monthly income	0.0442	***	0.0806		0.0916	***	0.0299		0.0168	*	0.0662	
Weekly working hours	−0.0124		−0.0952		−0.0344	*	0.0176		−0.0016		−0.1920	
Non-regular worker	−0.0048		0.0361		0.0063		0.1025	**	−0.0010		−0.0272	
Job loss experience	0.0137	**	0.0015		0.0145	**	−0.0313	*	0.0035		0.0394	*
Decrease in working hours	−0.0015		0.0109		−0.0021		−0.0173		−0.0011		0.0381	
Income decrease (own)	0.0110	**	−0.0102		0.0153	**	−0.0208		0.0058		−0.0109	
Income decrease (family members)	0.0119	**	−0.0146		0.0141		−0.0256		0.0095		0.0102	
Work from home not allowed	0.0111	**	−0.0214	*	0.0078		−0.0124		0.0137	*	−0.0318	
2nd wave	0.0006		−0.0070		0.0027		−0.0013		−0.0015		−0.0137	
3rd wave	0.0000		−0.0115		0.0006		−0.0457	**	−0.0011		0.0245	
4th wave	0.0002		0.0185		−0.0023		0.0440	**	0.0025		−0.0121	

Notes: Effects of firm size and region were estimated but are not included in this table. *** *p* < 0.01, ** *p* < 0.05, * *p* < 0.1.

## Data Availability

The data presented in this study are available on request from the corresponding author. The data are not publicly available due to ethical issues.

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
