# Peer review of "Gender Gap in Mental Health during the COVID-19 Pandemic in South Korea: A Decomposition Analysis"

_ijerph, 2023, doi:10.3390/ijerph20032250_

Round 1
Reviewer 1 Report
This study investigates gender differences in the relationship between COVID-19 and mental health in Korea. This critical topic warrants further investigation. This study has many strengths and is a pleasure to read. Nevertheless, I have several questions and concerns that the authors should consider.
First, it would be helpful to include a sentence in the abstract as to why the authors may expect gender differences in the link between COVID-19 and mental health. This additional sentence would better ground their investigation.
Second, if space permits, it would be helpful to add a bit more on why the context in Korea may be different from other countries. Specifically, if the authors could note how the statistics from other contexts are different from the Korean context in terms of gender and employment. Without some sort of comparison, it is hard to know how the Korean context may differ from other countries.
Third, it would be helpful to have additional information on the data collection method. Was this a random sample? How were respondents recruited to take the online survey?
Fourth, it would be helpful for the authors to explain the reasoning for the decomposition method in more simplistic terms, as many readers are likely to be unfamiliar with this method. An additional sentence or two explaining the advantages of this analysis would address this issue.
Finally, it would be helpful to add a few sentences in the frontend of the paper as to why the authors may expect the gender differences to vary by age. This would better prepare the reader for the analysis focusing on age differences in the links between gender, COVID-19, and mental health.
Author Response
Response to Reviewer 1 Comments
We appreciate the time and effort that you dedicated to providing feedback on ourmanuscript and are grateful for the insightful comments on our paper. The changes we have made aremarked up using the “track changes” function within the manuscript, and our responses to reviewer’s comments are below, in blue.
Point 1: First, it would be helpful to include a sentence in the abstract as to why the authors may expect gender differences in the link between COVID-19 and mental health. This additional sentence would better ground their investigation.
Author response 1: This is a good suggestion. We have added a sentence as to why we expect gender differences in the association between COVID-19 and mental health to the abstract. The sentence is: “The pandemic also may have affected men’s and women’s mental health differently. This study examined whether the pandemic had stronger adverse effects on women’s mental health than on that of men given that the decline of the labor market was greater for women than for men.”
Point 2: Second, if space permits, it would be helpful to add a bit more on why the context in Korea may be different from other countries. Specifically, if the authors could note how the statistics from other contexts are different from the Korean context in terms of gender and employment. Without some sort of comparison, it is hard to know how the Korean context may differ from other countries.
Author response 2: Thank you for this suggestion. We have added several sentences about how Korea is differenct from other countries in terms of gender and employment to the manuscript as follows on page 3. “Second, the gender gap in employment is larger and traditional gender role attitudes are stronger than those of other OECD countries; women work disproportionately more in service and sales jobs and spend much more time on household care than men. It is postulated that these differences worsened women’s mental health during the pandemic compared to those of women in countries with a smaller gender disparity.”
Point 3: Third, it would be helpful to have additional information on the data collection method. Was this a random sample? How were respondents recruited to take the online survey?
Author response 3: We have added a sentence explaining data collection method as follows on page 3. “The sample was selected using proportional distribution random sampling based on the proportion of the employed population from the Economically Active Population Survey conducted by Statistics Korea.” Thanks for the suggestion.
Point 4: Fourth, it would be helpful for the authors to explain the reasoning for the decomposition method in more simplistic terms, as many readers are likely to be unfamiliar with this method. An additional sentence or two explaining the advantages of this analysis would address this issue.
Author response 4: As suggested by the reviewer, we have added a couple of sentences explaining the B-O method as follows on page 4. “This method is used to determine the amount of group difference in outcome variable due to observable personal characteristics compared to that due to structural and unknown factors. As such, the B-O decomposition method has the advantage of not only specifying variables that contribute to the difference, but also identifying the difference explained by structural characteristics which is regarded as discrimination. The B-O decomposition is a special case of the statistical method to quantify the relative contribution of each mediator in X (e.g., labor market experiences) to the total effect of a focal predictor Z (e.g., gender) on the outcome variable Y (e.g., mental health) [33, 34]. Unlike the conventional approach in mediation analysis, which adds covariates sequentially into the regression, the B-O decomposition estimates the contributions of all potential mediators in a model at the same time. It does not depend on the assumption that confounders and mediators are not correlated, and its results are not affected by the ordering of variables. For a detailed description of the method the reader is referred to [32,38]” We appreciate the suggestion.
Point 5: Finally, it would be helpful to add a few sentences in the frontend of the paper as to why the authors may expect the gender differences to vary by age. This would better prepare the reader for the analysis focusing on age differences in the links between gender, COVID-19, and mental health.
Author response 5: Thank you for this suggestion. We have added a sentence explaining how we expect the gender difference to vary by age on page 3. The added sentence is: “Also, we expect that there was a significant age effect on this impact since women’s burden of housework might be greatest in their 30s and 40s when they have young children, while men did not demonstrate such age variation.”

Reviewer 2 Report
Intro
This is a good opening. You mention on page 3 that there are differences in household chores by gender. You go on to talk about how this gender divide has only increased during the pandemic. But I don’t see this discussed in the method section. While this is interesting, I think more time spent on why mental health disparities may arise due to changes in socioeconomic status or discussing how household work better fits within the changes in labor activity may be more useful? This might fit better higher up as well.
Methods
This looks good. The only question I would have is: does including weekly working hours as a control create a moderating / mediating affect? I ask because decreased hours is an independent variable. I can see these two variables being highly correlated with each other.
As I am not familiar with Oaxaca and Ransom or Jann’s discussion on decomposition, I would suggest an extra sentence or two explaining this to readers who are unfamiliar with this methodology. This may be especially important since you should justify why you used it for OLS when it is more appropriate for an ordered logit or probit model.
Results
These figures look great. Looking at Table 2: Interesting that the only age group to show greater distress was those in their 30s.
Discussion
I would like to see a bit more on how we might contextualize these findings in the overall COVID and mental health narrative. You briefly mention on page 10 why you found different results from studies out of the UK and Germany. How might South Korea compare to other high income nations such as the US, Canada, or Australia. Also, South Korea did relatively well with the pandemic compared to the rest of the world. The had only 24,696 coronavirus-related deaths reported in the country since the pandemic began. They also have one of the highest vaccination rates in the world (with 122% of their population vaccinated according to Reuters link below). Would you expect this change in mental health to be temporary as the country quickly got the virus under control? Or do you expect long-term gaps to persist based on previous understandings? Also, what directions for future research might be implied for all of this – whether we examine similar measures in other nations or what it means for South Korea specifically?
https://www.reuters.com/graphics/world-coronavirus-tracker-and-maps/countries-and-territories/south-korea/
Overall, I really liked this paper. I think that it contributes to our understanding of the impacts of COVID and with just a few tweaks, could really add to the literature currently being built.
Author Response
Response to Reviewer 2 Comments
We appreciate the time and effort that you dedicated to providing feedback on ourmanuscript and are grateful for the insightful comments on our paper. The changes we have made are marked up using the “track changes” function within the manuscript, and our responses to reviewer’s comments are below, in blue.
Point 1: Intro
This is a good opening. You mention on page 3 that there are differences in household chores by gender. You go on to talk about how this gender divide has only increased during the pandemic. But I don’t see this discussed in the method section. While this is interesting, I think more time spent on why mental health disparities may arise due to changes in socioeconomic status or discussing how household work better fits within the changes in labor activity may be more useful? This might fit better higher up as well.
Author response 1: Thanks for pointing these issues out. First, we have added a discussion on the gender divide in the method section on page 4, as suggested by reviewer as follows: “Three regression models were analyzed. In a pooled model, we estimated the effects of independent variables on mental health by including all the individuals, and the same analysis was applied to two models constructed separately for women and men to examine how the independent variables act differently in these two groups.”
Second, we have a couple of sentences about how Korea is differenct from other countries in terms of gender and employment to address the issue that the reviewer raised to the manuscript as follows on page 3. “Second, the gender gap in employment is larger and traditional gender role attitudes are stronger than those of other OECD countries; women work disproportionately more in service and sales jobs and spend much more time on household care than men. It is postulated that these differences worsened women’s mental health during the pandemic compared to those of women in countries with a smaller gender disparity.”
Point 2: Methods
This looks good. The only question I would have is: does including weekly working hours as a control create a moderating / mediating affect? I ask because decreased hours is an independent variable. I can see these two variables being highly correlated with each other.
Author response 2: This is an important question. We have conducted the same regressions analyses without a variable ‘weekly working hours’ to check its mediating effect. The results were basically the same; a notable mediating effect was not shown. Also, although ‘weekly working hours’ is significantly correlated with many key variables used in our study, a VIF test did not show an indication of multicollinearity. We have added following sentences on page 8;“In addition, we conducted the same regression analyses using models without weekly working hours to assess its mediating effect. However, there was no significant difference in the results, indicating no notable mediating effect of weekly working hours. Also, although weekly working hours are significantly correlated with some key variables used in our study, a VIF test did not show an indication of multicollinearity.”
Point 3: As I am not familiar with Oaxaca and Ransom or Jann’s discussion on decomposition, I would suggest an extra sentence or two explaining this to readers who are unfamiliar with this methodology. This may be especially important since you should justify why you used it for OLS when it is more appropriate for an ordered logit or probit model.
Author response 3: We have added a couple of sentences explaining what advantages of using the B-O decomposition method are on page 5. “This method is used to determine the amount of group difference in outcome variable due to observable personal characteristics compared to that due to structural and unknown factors. As such, the B-O decomposition method has the advantage of not only specifying variables that contribute to the difference, but also identifying the difference explained by structural characteristics which is regarded as discrimination. The B-O decomposition is a special case of the statistical method to quantify the relative contribution of each mediator in X (e.g., labor market experiences) to the total effect of a focal predictor Y (e.g., gender) on the outcome variable Z (e.g., mental health) [33, 34]. Unlike the conventional approach in mediation analysis, which adds covariates sequentially into the regression, the B-O decomposition estimates the contributions of all potential mediators in a model at the same time. It does not depend on the assumption that confounders and mediators are not correlated, and its results are not affected by the ordering of variables. For a detailed description of the method the reader is referred to [32,38]” We appreciate the suggestion.
Point 4: Results
These figures look great. Looking at Table 2: Interesting that the only age group to show greater distress was those in their 30s.
Author response 4: Thank you!
Point 5: Discussion
I would like to see a bit more on how we might contextualize these findings in the overall COVID and mental health narrative. You briefly mention on page 10 why you found different results from studies out of the UK and Germany. How might South Korea compare to other high income nations such as the US, Canada, or Australia. Also, South Korea did relatively well with the pandemic compared to the rest of the world. The had only 24,696 coronavirus-related deaths reported in the country since the pandemic began. They also have one of the highest vaccination rates in the world (with 122% of their population vaccinated according to Reuters link below). Would you expect this change in mental health to be temporary as the country quickly got the virus under control? Or do you expect long term gaps to persist based on previous understandings? Also, what directions for future research might be implied for all of this – whether we examine similar measures in other nations or what it means for South Korea specifically?
Author response 5: We think this is an excellent suggestion. We have added a short paragraph as suggested by reviewer on page 12 as follows.
“Although many countries are starting to return to pre-pandemic activities, the global economy remains strained. South Korea has been considered to respond better to the COVID-19 pandemic than other high-income countries in terms of the number of deaths, rate of vaccination, etc. However, it is uncertain whether Korean society also will recover from the pandemic faster than other countries. Despite the recent increase in gender role flexibility in the labor market and within families, gender issues persist in many areas, and family-friendly social systems are not well established in South Korea. As long as the current structural discrimination persists, the higher social cost of women will continue.”
